# Data-driven Estimation of Sinusoid Frequencies

**Gautier Izacard**
Ecole Polytechnique
gautier.izacard@polytechnique.edu

**Sreyas Mohan**
Center for Data Science
New York University
sm7582@nyu.edu

**Carlos Fernandez-Granda**
Courant Institute of Mathematical Sciences,
and Center for Data Science
New York University
cfgranda@cims.nyu.edu

## Abstract

Frequency estimation is a fundamental problem in signal processing, with applications in radar imaging, underwater acoustics, seismic imaging, and spectroscopy. The goal is to estimate the frequency of each component in a multisinusoidal signal from a finite number of noisy samples. A recent machine-learning approach uses a neural network to output a learned representation with local maxima at the position of the frequency estimates. In this work, we propose a novel neural-network architecture that produces a significantly more accurate representation, and combine it with an additional neural-network module trained to detect the number of frequencies. This yields a fast, fully-automatic method for frequency estimation that achieves state-of-the-art results. In particular, it outperforms existing techniques by a substantial margin at medium-to-high noise levels.

## 1 Introduction

### 1.1 Estimation of sinusoid frequencies

Estimating the frequencies of multisinusoidal signals from a finite number of samples is a fundamental problem in signal processing. Examples of applications include underwater acoustics [2], seismic imaging [5], target identification [3, 11], digital filter design [37], nuclear-magnetic-resonance spectroscopy [43], and power electronics [27]. In radar and sonar systems, the frequencies encode the direction of electromagnetic or acoustic waves arriving at a linear array of antennae or microphones [26].

In signal processing, multisinusoidal signals are usually represented as linear combinations of complex exponentials,

$$S(t) := \sum_{j=1}^{m} a_j \exp(i 2\pi f_j t) = \mathrm{Re}(a_j) \cos(2\pi f_j t) + i \, \mathrm{Im}(a_j) \sin(2\pi f_j t). \tag{1}$$

where the unknown amplitudes $a \in \mathbb{C}^m$ encode the magnitude and phase of the different sinusoidal components, and $t$ denotes time. The frequencies $f_1, \ldots, f_m$ quantify the oscillation rate of each component. The goal of frequency estimation is to determine their values from noisy samples of the signal $S$.

Without loss of generality, let us assume that the true frequencies belong to the unit interval, i.e. $0 \leq f_j \leq 1, 1 \leq j \leq m$. By the Sampling Theorem [25, 30, 35] the signal in equation 1 is completely

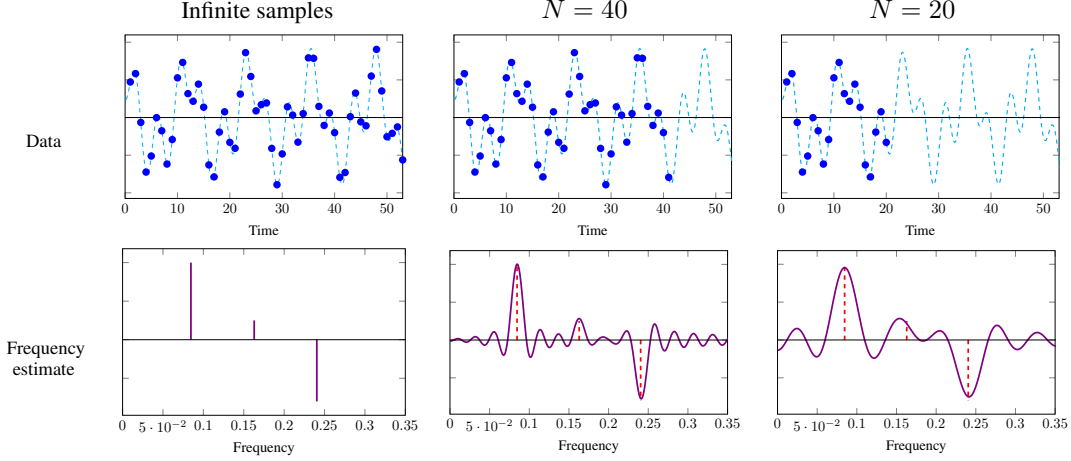

Figure 1: Illustration of the frequency-estimation problem. A multisinusoidal signal given by equation 1 (dashed blue line) and its Nyquist-rate samples (blue circles) are depicted on the top row. The bottom row shows that the resolution of the frequency estimate obtained by computing the discrete-time Fourier transform from $N$ samples decreases as we reduce $N$. The signal is real-valued, so its Fourier transform is even; only half of it is shown.

determined by samples measured at a unit rate[1]: $\ldots, S(-1), S(0), S(1), S(2), \ldots$ Computing the discrete-time Fourier transform from such samples recovers the frequencies exactly (intuitively, the discretized sinusoids form an orthonormal basis, see e.g. [31]). However this requires an *infinite* number of measurements, which is not an option in most practical situations.

In practice, the frequencies must be estimated from a *finite* number of measurements corrupted by noise. Here we study a popular measurement model, where the signal is sampled at a unit rate,

$$y_k := S(k) + z_k, \qquad 1 \le k \le N, \tag{2}$$

and the noise $z_1, \ldots, z_N$ is additive. Limiting the number of samples is equivalent to multiplying the signal by a rectangular window of length $N$. In the frequency domain, this corresponds to a convolution with a kernel (called a discrete sinc or Dirichlet kernel) of width $1/N$ that blurs the frequency information, limiting its resolution as illustrated in Figure 1 (see Section 1 in the Supplementary Material for a more detailed explanation). Because of this, the frequency-estimation problem is often known as *spectral super-resolution* in the literature (in signal processing, the spectrum of a signal refers to its frequency representation).

## 1.2 State of the art

A natural way to perform frequency estimation from data following the model in equation 2 is to compute the magnitude of their discrete-time Fourier transform. This is a linear estimation method known as the *periodogram* in the signal-processing literature [39]. As illustrated by Figure 1 and explained in more detail in Section 1 of the Supplementary Material, the periodogram yields a superposition of kernels centered at the true frequencies. The interference produced by the side lobes of the kernel complicates finding the locations precisely[2] (see for example the middle spike in Figure 1 for $N = 20$). The periodogram consequently does not recover the true frequencies exactly, even if there is no noise in the data. However, it is a popular technique that often outperforms more sophisticated methods when the noise level is high.

The sample covariance matrix of the data in equation 2 is low rank [39]. This insight can be exploited to perform frequency estimation by performing an eigendecomposition of the matrix, a method known as MUltiple SIgnal Classification (MUSIC) [4, 34]. The approach is related to Prony's method [32, 42]. In a similar spirit, matrix-pencil techniques extract the frequencies by forming a

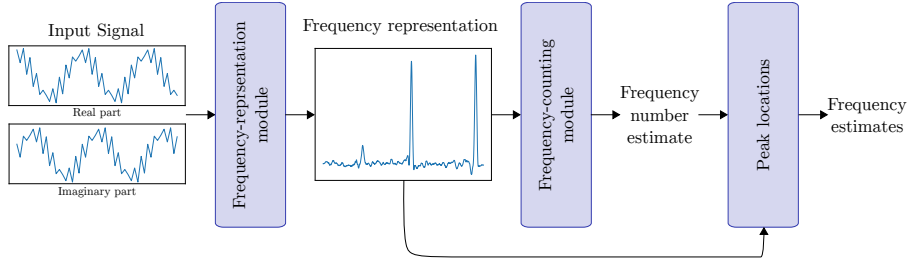

Figure 2: Architecture of the DeepFreq method.

matrix pencil before computing the eigendecomposition of the sample covariance matrix [21, 33]. We refer to [38] for an exhaustive list of related methods. Eigendecomposition-based methods are very accurate at low noise levels [28, 29], and provably achieve exact recovery of the frequencies in the absence of noise, but their performance degrades significantly as the signal-to-noise ratio decreases.

The periodogram and eigendecomposition-based methods assume prior knowledge of the number of frequencies to be estimated, which is usually not available in practice. Classical approaches to estimate the number of frequencies use information-theoretic criteria such as the Akaike information criterion (AIC) [44] or minimum description length (MDL) [45]. Both methods minimize a criterion based on maximum likelihood that involves the eigenvalues of the sample covariance matrix. An alternative technique known as the second-order statistic of eigenvalues (SORTE) [20, 17] produces an estimate of the number of frequencies based on the gap between the eigenvalues of the sample covariance matrix.

Variational techniques are based on an interpretation of frequency estimation as a sparse-recovery problem. Sparse solutions are obtained by minimizing a continuous counterpart of the $\ell_1$ norm [10, 40, 15]. The approach has been extended to settings with missing data [41], outliers [16], and varying noise levels [8]. As in the case of eigendecomposition-based methods, these techniques are known to be robust at low noise levels [9, 13, 40, 14], but exhibit a deteriorating empirical performance as the noise level increases. An important drawback of this methodology is the computational cost of solving the optimization problem, which is formulated as a semidefinite program or as a quadratic program in very high dimensions.

Very recently, the authors of [23] propose a learning-based approach to frequency estimation based on a deep neural network. The method is shown to be competitive with the periodogram and eigendecomposition-based methods for a range of noise levels, but requires an estimate of the number of sinusoidal components as an input. Other recent works apply deep learning to related inverse problems, including sparse recovery [47, 19], point-source superresolution [7], and acoustic source localization [1, 46, 12].

## 1.3 Contributions

This work introduces a novel deep-learning framework to perform frequency estimation from data corrupted by noise of unknown variance. The approach is inspired by the learning-based method in Ref. [23], which generates a frequency representation that can be used to perform estimation if the number of true frequencies is known. In this work, we propose a novel neural-network architecture that produces a significantly more accurate frequency representation, and combine it with an additional neural-network module trained to estimate the number of frequencies. This yields a fast, fully-automatic method for frequency estimation that achieves state-of-the-art results. The approach outperforms existing techniques by a substantial margin at medium-to-high noise levels. Our results showcase an area of great potential impact for machine-learning methodology: problems with accurate physical models where model-based methodology breaks down due to stochastic perturbations that can be simulated accurately. The code used to train and evaluate our models is available online at `https://github.com/sreyas-mohan/DeepFreq`.

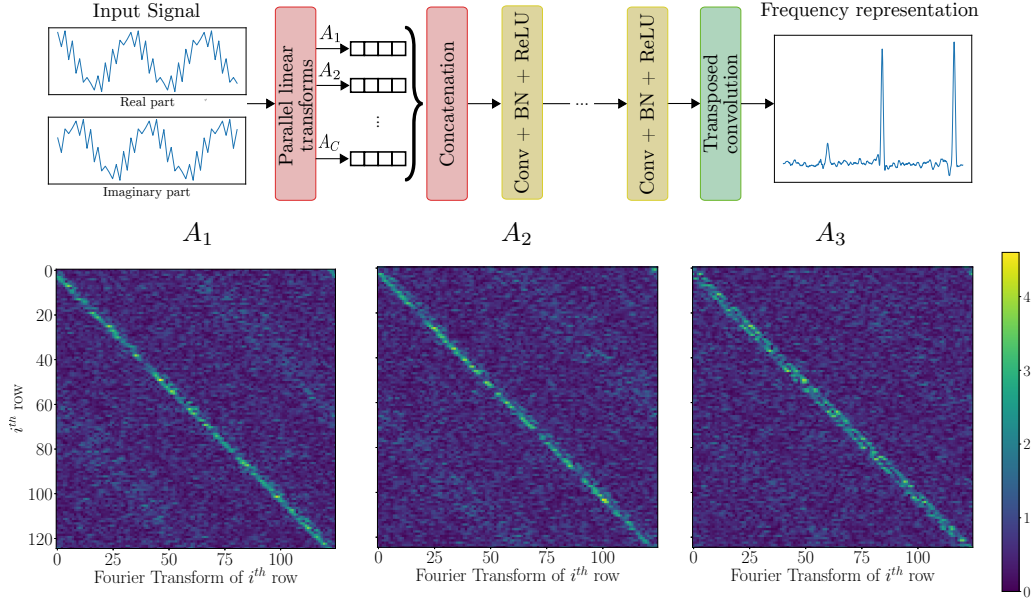

Figure 3: *Top*: Architecture of the DeepFreq frequency-representation module described in Section 2.2. *Bottom*: Heat maps showing the magnitudes of the Fourier transform of the rows of the matrices associated to three of the channels in the first layer of the encoder of the frequency-representation module. The diagonal pattern indicates that each channel computes a Fourier-like transformation. Note that the frequencies are ordered automatically. The reason is that after the first layer the network is convolutional and has a reduced field of view. In order to produce an accurate frequency representation at the output, the first layer needs to order the relevant frequency information so that it can be propagated by the convolutional layers.

## 2 Methodology

### 2.1 Overview

Most existing techniques for frequency estimation build continuous frequency representations of the observed data, as opposed to estimating the frequencies directly. In the case of the periodogram, the representation is just the discrete-time Fourier transform of the measurements. In the case of eigendecomposition-based methods, a different representation– known as the pseudo-spectrum– is computed using a subset of the eigenvectors of the sample covariance matrix of the data. One can show that in the absence of noise, the peaks of the pseudo-spectrum are located exactly at the locations of the true frequencies. For noisy data, the hope is that the perturbation does not vary the locations too much. In the case of variational methods, yet another representation is obtained from the solution to the dual of the sparsity-promoting convex program [10]. In this case, the frequencies are estimated by locating local maxima that have magnitude close to one.

Recently, the authors of [23] propose generating a frequency representation in a data-driven manner, training a neural network called the PSnet to produce it directly from the measurements. Frequency estimation is then carried out by finding the peaks of the representation. The authors show that the approach is more effective than using a deep-learning model to directly output the frequency values. Building upon the idea of learned frequency representations, we propose an improved version of the PSnet and combine it with an additional neural network that performs automatic estimation of the number of frequencies. Figure 2 shows a diagram of the architecture. First, the data are fed through a module that produces a frequency representation. Then, the representation is fed into a second *frequency-counting* module that outputs an estimate of the number of sinusoidal components $\widehat{m}$. Finally, the frequency estimates are computed by locating the $\widehat{m}$ highest maxima in the frequency representation. We call this method DeepFreq. Sections 2.2 and 2.3 describe the proposed architectures for the frequency-representation and frequency-counting modules respectively.

### 2.2 Frequency-representation module

Building upon the methodology proposed in Ref. [23], we implement the frequency-representation module as a feedforward deep neural network. Given a set of true frequencies $f_1, \ldots, f_m$, we define a ground-truth frequency

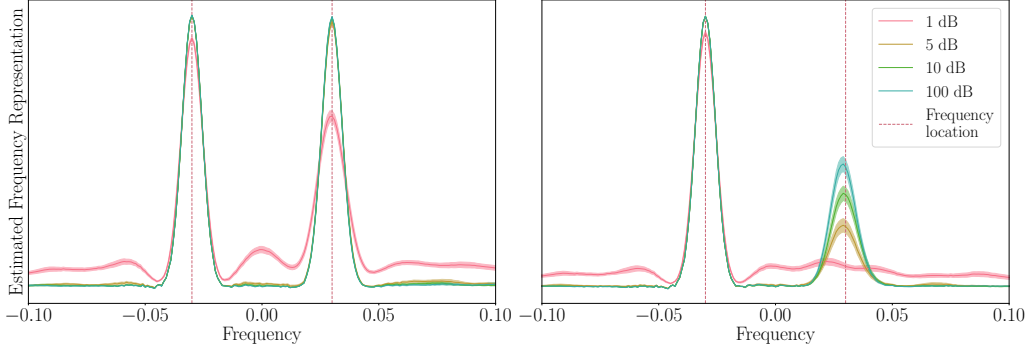

Figure 4: Frequency representation learned by DeepFreq for data generated from a signal with two sinusoidal components. The amplitude of the first component has magnitude equal to one. The second component has magnitude equal to 0.5 (left) and 0.1 (right). For four different signal-to-noise ratios, the representation is averaged over 100 signals with random phases and different noise realizations. The error bars represent standard error.

representation *FR* as a superposition of narrow Gaussian kernels $K : \mathbb{R} \to \mathbb{R}$ centered at each frequency

$$\text{FR}\,(u) := \sum_{j=1}^{m} K\,(u - f_j)\,. \tag{3}$$

*FR* is a smooth function that has sharp peaks at the location of the true frequencies, and decays rapidly away from them. Note that amplitude information is *not* encoded in *FR*; each shifted kernel has the same amplitude. The neural network is calibrated to output an approximation to *FR* from $N$ noisy, low-resolution data given by the model in equation 2. This is achieved by minimizing a training loss that penalizes the square $\ell_2$-norm approximation error between the output and the true *FR* function over a fine grid for a database of examples.

Figure 3 shows the proposed architecture for the frequency-representation module. The overall structure is similar to the PSnet architecture from Ref. [23]. First, a linear encoder maps the input data to an intermediate feature space. Then, the features are processed by a series of convolutional layers with localized filters of length 3 and batch normalization [22], interleaved with ReLUs. The dimension of the input is preserved using circular padding. Finally, a decoder produces the *FR* estimate applying a transposed convolution (in the PSnet a fully connected layer is used instead). If the data are complex-valued, the real and imaginary parts are processed as pairs of real numbers.

The main difference between our proposed architecture and the PSnet is the encoder. Intuitively, the encoder learns a Fourier-like transformation that concentrates frequency information locally so that it can be processed by the convolutional filters in the subsequent layer. The PSnet uses a single linear map to implement the transformation: for an input $y \in \mathbb{C}^N$ the output of the encoder is $Ay$, where $A$ is a fixed $M \times N$ matrix and $M > N$. We propose to instead use multiple separate linear maps. The output of the DeepFreq encoder can be represented by a feature matrix

$$\begin{bmatrix} A_1 y & A_2 y & \cdots & A_C y \end{bmatrix}, \tag{4}$$

where each $A_i$, $1 \le i \le C$, is a fixed $M \times N$ matrix. The $C$ columns can be interpreted as different channels, which extract complementary features from the input. The filters in the next layer of the architecture combine the information of all channels, while acting convolutionally on the columns of the feature matrix. Visualizing the Fourier transform of the rows of $A_1, \ldots, A_C$ for a trained DeepFreq network reveals that each of the channels implement similar, yet different, Fourier-like transformations: the rows are approximately sinusoidal, with frequencies that are ordered sequentially (see Figure 3). This provides a rich set of frequency features to the convolutional layers, which boosts the performance of the frequency-representation module with respect to the PSnet (see Section 3.2).

## 2.3 Frequency-counting module

Figure 4 shows the output of the frequency-representation module for a simple signal with two sinusoidal components. When one of the components has small amplitude and the data are noisy, the representation may still detect the frequency, but the magnitude of the corresponding peak decreases. In addition, spurious local maxima may appear due to the stochastic fluctuations in the data. In order to perform estimation by locating maxima in the learned representation, it is necessary to first decide how many components to look for. This is a pervasive problem in frequency estimation, which is also an issue for traditional methods. Many published works assume that the number of components is known beforehand (including [23]), but this is often not the case

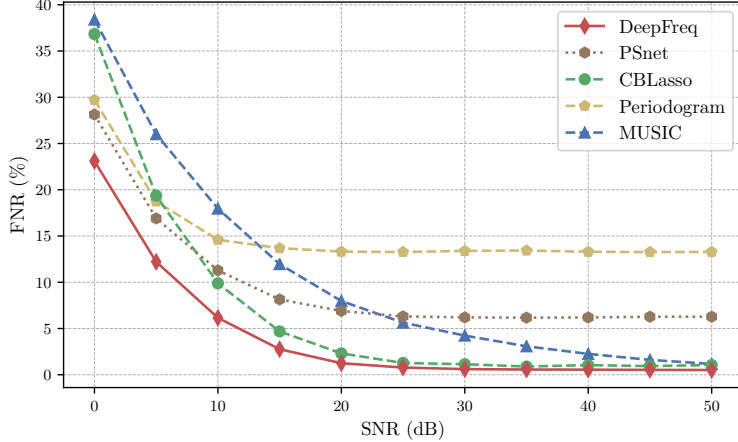

Figure 5: False negative rate of DeepFreq compared to other methodologies. DeepFreq outperforms all other methods, including PSnet. Only CBLasso at high signal-to-noise ratios exhibits similar performance. The experiment is described in Section 3.2.

in many practical applications. In this section we describe a frequency-counting module designed to estimate the number of sinusoidal components automatically.

We propose to implement the frequency-counting module using a neural network. The network is trained to extract the number of components from the output of the frequency-representation module in Section 2.2. The representation produced by the module concentrates the frequency information locally, which makes it easier to count the number of components. Patterns indicating the presence of true frequencies can be expected to be invariant to translations as long as the noise is not structured in the frequency domain. We exploit this insight by applying a convolutional architecture to count the frequencies. An initial 1D strided convolutional layer with a wide kernel is followed by several convolutional blocks with localized filters. The final layer is fully connected. It outputs a single real number, which is rounded to the nearest integer to produce the count estimate. The counting-module is calibrated on a training dataset containing instances of *FR* functions produced by the frequency-representation module. Note that the frequency-representation and frequency-counting modules are trained separately. The loss function is given by the squared $\ell_2$ norm difference between the count estimate and the true number of sinusoidal components. Section 3.3 shows that our approach clearly outperforms eigendecomposition-based methods at medium-to-high noise levels.

# 3 Computational experiments

## 3.1 Experimental design

To validate our approach we simulate data according to the signal model in equation 1 and the measurement model in equation 2 for $N := 50$. The data generation process is the following:

1. The number of components $m$ in each signal is chosen uniformly at random between 1 and 10.

2. The frequency values $f_1, \ldots, f_m$ are generated so that the minimum separation between them is greater or equal to $1/N$. The minimum separation governs the difficulty of locating the differences. Under $2/N$ the problem is very challenging and under $1/N$ it is almost impossible (we refer the reader to [29, 36, 10] for an in-depth analysis of this phenomenon). The separation between the frequencies is set to equal $1/N + |w|$, where $w$ is a Gaussian random variable with standard deviation equal to $2.5/N$.

3. The coefficients $a_j$, $1 \le j \le m$, are given by $a_j := (0.1 + |w_j|) \, e^{i\theta_j}$, where $w_j$ is sampled from a standard Gaussian distribution and the phase $\theta_j$ is uniform in $[0, 2\pi]$. The minimum possible amplitude also governs the difficulty of the problem. We fix it to 0.1.

4. The noise level varies in a certain range, and is considered unknown. For each noise realization, we first sample the noise level $\sigma$ uniformly in the interval $[0, 1]$. Then we generate $N$ i.i.d. standard Gaussian samples. Finally, we scale the noise so that the ratio between the $\ell_2$ norm of the noise and the signal equals $\sigma$. This yields a range of signal-to-noise ratios (SNR) between 0 dB and $\infty$.

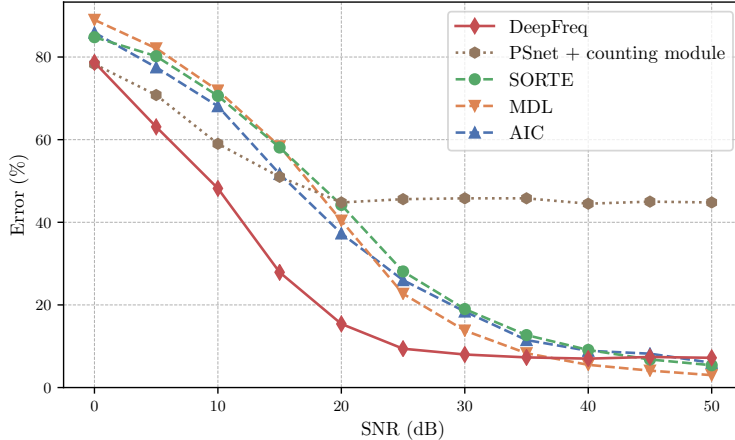

Figure 6: Average error of the DeepFreq frequency-counting module, the DeepFreq frequency-counting module trained with the output of the PSnet, and three representative eigendecomposition-based methods for the experiment described in Section 3.3.

## 3.2 Frequency representation

As mentioned in Section 2, most existing methods for frequency estimation construct frequency representations. Here we compare these representations to the one learned by DeepFreq, in a setting where the noise level in the data is unknown. We consider four representative methods: the periodogram [39], MUSIC [4, 34], a variational method known as the concomitant Beurling lasso (CBLasso) [8], and the PSnet method in [23].

The architecture of the DeepFreq frequency-representation module follows the description in Section 2.2. We fix the standard deviation of the Gaussian filter in the representation to $0.3/N$. We train a single model for the whole range of noise levels. The number of channels $C$ in the encoder is set to 64. The output dimensionality $M$ of the encoder is set to 125. The number of intermediate convolutional layers is set to 20. The width of the filter in the transposed convolution in the decoder is set to 25 with a stride of 8 in order to obtain a discretization of the representation on a grid of size $10^3$. We build the training set generating $2 \cdot 10^5$ clean signals. During training, new noise realizations are added at each epoch. The training loss is minimized using the Adam optimizer [24] with a starting learning rate of $3 \cdot 10^{-4}$. The same training procedure is used to train the PSnet network.

We evaluate the different methods on a test set where the clean signal samples follow the model in Section 3.1. For each noise level, we generate $10^3$ signals, which are different from the ones in the training set. We assume that the true number of sinusoidal components $m$ is known. The frequency estimates $\hat{f}_1, \ldots, \hat{f}_m$ are obtained by locating the highest $m$ maxima of the frequency representations constructed by the different methods from the noisy data. The representations are evaluated on a fine grid with $10^3$ points. The accuracy of the estimate is measured by computing the false negative rate *FNR*. The *FNR* is defined as the number of true frequencies that are undetected, meaning that there is no estimated frequency within a radius of $(2N)^{-1}$ (recall that the minimum separation is $1/N$).

Figure 5 shows the results. The DeepFreq frequency-representation module outperforms all other methods at low-to-middle SNRs, and is only matched by CBLasso at high SNRs. In particular, it outperforms the PSnet by between 4% and 7% over the whole range of noise levels. It is worth noting, that CBLasso is extremely slow: its average running time is 1.71 seconds. The DeepFreq module is two orders of magnitude faster (42 milliseconds)[3].

## 3.3 Frequency counting

In this section we report the performance of the DeepFreq frequency-counting module. To the best of our knowledge, the only existing techniques to estimate the number of sinusoidal components rely on an eigende-composition of the sample covariance matrix of the data. We compare to three of the most popular examples: AIC [44], MDL [45] and SORTE [20].

The architecture of the module is convolutional with a final fully-connected layer, as described in 2.3. The initial layer contains 16 filters of size 25 with a stride of 5, which downsample the input into features vectors of length 200. We set the number of subsequent convolutional layers to 20, each containing 16 filters of size 3. We generate training data by feeding the training data described in Section 3.2 through a DeepFreq frequency-representation

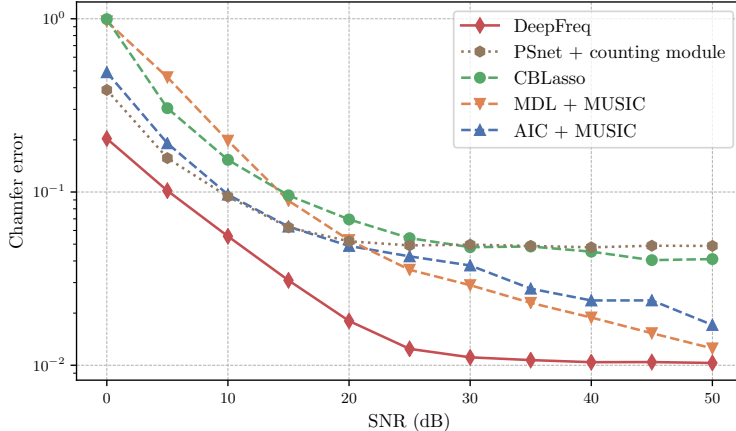

Figure 7: Frequency-estimation performance of DeepFreq compared to other methodologies. Standard error bars for the DeepFreq method are shown in Section 2 of the Supplementary material. The experiment is described in Section 3.4.

module with fixed, calibrated weights. The training loss is minimized using the Adam optimizer [24]. Figure 6 shows the fraction of signals in the test set for which the number of components is not estimated correctly for different methodologies (the test data is generated as in Section 3.2). The DeepFreq frequency-counting module clearly outperforms the eigendecomposition-based methods except at very high signal-to-noise ratios. A natural question to ask is how DeepFreq compares to a model using our counting module combined with the PSnet. To investigate this, we train the proposed frequency-counting module using the representation produced by PSnet. As shown in Figure 6 replacing the DeepFreq representation by the PSnet representation results in a significant decrease in performance. This suggests that the performance of the counting module is highly dependent on the quality of the frequency representation provided as input.

## 3.4 Frequency estimation

In this section we evaluate the frequency-estimation performance of DeepFreq in a realistic setting where both the noise level and the number of sinusoidal components are unknown. The DeepFreq modules are calibrated separately, as described in Sections 3.2 and 3.3. Training takes 11 hours on an NVIDIA P40. The test data are generated as described in Section 3.2. We compare our approach to an eigendecomposition-based procedure that combines MUSIC with AIC or MDL, the CBLasso (where frequencies are selected from the dual solution using a threshold calibrated with a validation dataset), and to a model combining the PSnet with our proposed frequency-counting module. We measure estimation accuracy by computing the Chamfer distance [6] between the $m$ true frequencies $f := (f_1, \ldots, f_m)$ and the $\widehat{m}$ estimates $\hat{f} := (\hat{f}_1, \ldots, \hat{f}_{\widehat{m}})$:

$$d(f, \hat{f}) = \sum_{f_i \in f} \min_{\hat{f}_j \in \hat{f}} \left| f_i - \hat{f}_j \right| + \sum_{\hat{f}_j \in \hat{f}} \min_{f_i \in f} \left| \hat{f}_j - f_i \right|. \tag{5}$$

Figure 7 shows the results. DeepFreq clearly outperforms the other methods over the whole range of noise levels.

## 4 Conclusion and future work

In this paper, we introduce a machine-learning framework for frequency estimation, which combines two neural-network modules calibrated with simulated data. The approach achieves state-of-the-art performance, is fully automatic, and can operate at varying (and unknown) signal-to-noise ratios. Our framework can be extended to other signal and noise models by modifying the training dataset accordingly. Our results illustrate an incipient shift of paradigm in modern signal processing, from model-based methods towards learning-based techniques. An interesting direction for future research is to design learning-based models capable of generating frequency representations that can be interpreted probabilistically in terms of the uncertainty of the estimate.

## Acknowledgements

C.F. was supported by NSF award DMS-1616340.

## Footnotes

[1]If we consider frequencies supported on an interval of length $\ell$, then the sampling rate must equal $\ell$.

[2]To alleviate the interference one can multiply the data with a smoothing window, but this enlarges the width of the blurring kernel and consequently reduces the resolution of the data in the frequency domain [18].

[3]Running times are measured on an Intel Core i5-6300HQ CPU.

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
