[Supplementary Material]

# Supplementary Material for Data-driven Estimation of Sinusoid Frequencies

## 1 Loss of resolution due to truncation in time

The loss of resolution caused by sampling over a finite interval becomes apparent when we compute the discrete-time Fourier transform (DTFT) of the truncated samples. The DTFT of $N$ samples of the multisinusoidal signal in equation 1, denoted by $S_N$, equals:

$$\mathrm{DTFT}(S_N)(f) := \sum_{k=1}^{N} S(k) \exp(-i2\pi k f) \tag{1}$$

$$= \sum_{k=1}^{N} \sum_{j=1}^{m} a_j \exp(i2\pi k f_j) \exp(-i2\pi k f) \tag{2}$$

$$= \sum_{j=1}^{m} a_j D_N (f - f_j), \qquad D_N (f) := \sum_{k=1}^{N} \exp(-i2\pi k f). \tag{3}$$

The kernel $D_N$ is called a discretized sinc or Dirichlet kernel in the literature. As $N \to \infty$ $D_N(f)$ converges to a Dirac measure, which provides infinite resolution: the DTFT of the signal consists of Dirac deltas centered exactly at the frequencies. For finite $N$ the DTFT of the samples is equal to the convolution between the DTFT of the signal and the kernel $D_N$. This is illustrated with a simple example in Figure 1. The width of the main lobe of $D_N$ equals $1/N$, which can be interpreted as the frequency resolution of the samples.

Figure 1: Illustration of the frequency-estimation problem. Truncation in the time domain is equivalent to convolving with a blurring kernel in the frequency domain, which limits the frequency resolution of the data.

## 2 Standard error bars for frequency-estimation results

Figure 2 shows the frequency-estimation results of DeepFreq with standard error bars for the experiment described in Section 3.4 of the main paper.

Figure 2: Frequency-estimation results of DeepFreq with standard error bars.