[Reviews · NeurIPS 2019]

Reviewer 1



**Originality** The proposed framework is original and interesting but it was recently introduced by authors of (23), so it is not a contribution of this paper. It's an incremental work from (23) which provides enough improvement, in particular the method to estimate the number of component. **Quality** The paper is of good quality. There is no theoretical results. The claims are well supported by numerical experiments and performances are compared to standard frequency estimation methods. The described method is completed and ready to use. Authors give an honest evaluation of their work in term of performances. Yet, when using neural network one important aspect is the ressources required for training which can differ significantly from ressources required by non-deepnet-based methods. For that reason, when authors mention running time (l217-218), I think it's a bit unfair to other methods. For a fairer comparison and considering that a network should be trained on artificial data before being applied to a specific dataset, authors should mention training time and energy consumption (or CPU number/GPU type). **Clarity** The paper is very well written and well organized. It is easy to read and it seems easy to reproduce. Minor correction: the function FR shouldn't be in italic in the text. **Significance** The results are out-performing state-of-the art methods. The main idea (learning-based techniques on artificial data) could be very useful to the signal processing/data analysis community but it was proposed in the recent paper (23). === Authors Feedback === Other reviewers and authors' response convinced me that this work was a significant improvement of existing reference (23) which was my main concern. Therefore, I increased the score.

Reviewer 2



The paper is well written and very clear. Novelties over state-of-the-art are well described and the authors do not oversell their method, by always referring to existing approaches. Performances of the method in this context are clearly demonstrated but I would have liked a deeper investigation on some points : - One of the novelty of the approach is the component counting module : I would have liked to see the influence of this module in terms of performances : what if this module is removed ? Are the improvement due to this module or to the change in the encoder architecture ? What if the number of components is misevaluated ? Does it degrade the results ? - Same questions about the noise level : what if this parameter is misevaluated ? Section 3.4 gives some insights about this, but I think that the fact that the method is fully automatic is an important point that would have needed a more detailed discussion. - How would the standard PSnet behave with an additional component counting module ? Would it achieve the same performances ? After rebuttal : I'm satisfied by the answers the authors gave on the precise influence of the changes they made over [23] : it was one of my main concerns and I find their arguments convincing. The new results clearly show that they have investigated the robustness and cared to make a fully automatic method, which is a nice addition to these algorithms that are often difficult to calibrate. The conception of the counting module is also not trivial, as described in the new simulation : adding a trivial counting module to PSNet does not allow to achieve the same performances.

Reviewer 3



This paper describes an improved approach for sinusoid frequency estimation combined with a new approach for sinusoid counting. It then puts them together and compares with several state of the art approaches to each problem. Experiments show that it outperforms them in both tasks on mixtures of 1-10 sinusoids of length 50 samples mixed with white Gaussian noise at SNRs between 0 and 50dB. Originality: While the frequency estimation encoder builds upon previous work, the first experiment shows that when given the proper number of sinusoids, it is able to select frequencies close to the true frequencies 4-8% absolute better than the PSNet method it is based on across SNR. Thus, it is original enough to provide better results. The originality is further enhanced by the proposed source counting module, which works better than the baselines it is compared to. Quality: The paper does a good job of making measured claims and supporting them with experimental verification. This is exemplified by its statement of contribution: "Our results showcase an area of great potential impact for machine-learning methodology: problems with accurate physical models where model-based methodology breaks down due to stochastic perturbations that can be simulated accurately." Clarity: The paper is very well written and easy to follow. The contributions are stated explicitly. The theory of sinusoid frequency estimation is described clearly and concisely. The highlights of a large related literature are summarized clearly. There is one issue that should be clarified, however. For the sinusoid counting task, the error metric is not described. This is an issue in the reader's ability to understand the training of the algorithm, but more importantly its evaluation in Figure 6. What is "Error" as listed on the y-axis? It seems that this is using a categorical loss where each number of sinusoids represents a different category. But this seems too strict and a regression approach might provide a cleaner training signal for the model. What is being done should at least be described clearly. Significance: sinusoid frequency estimation is an important problem in several fields. As described in the paper, these include, "radar imaging, underwater acoustics, seismic imaging, and spectroscopy." The ability to do this better than existing methods, including sinusoid counting as well, is significant. The experimental validation is not completely realistic, but it is in a challenging regime of a small number of samples and a high noise level where existing methods fail. Overall, this paper presents a significant advancement of the deep learning-ification of a classic problem in signal processing. Author feedback ---------------------- I have read the author feedback and found it to be quite convincing. The additional results provided using the proposed counting module with the existing PSNet architecture show that the improvements to the frequency estimator provide a significant boost in performance. The additional details and explanation about the counting module are helpful as well.

[Author Response · NeurIPS 2019]

We thank the reviewers for their time, and for their valuable feedback, which will improve the quality and clarity of the manuscript. We feel that we didn't sufficiently highlight the contributions of our methodology with respect to PSnet. PSnet provides a frequency estimate that can be used to perform frequency estimation only if the right number of components is known. Then it is competitive with traditional methods at high noise levels. In contrast, DeepFreq provides an improved frequency representation learned with a novel multilinear architecture, which is significantly superior to PSnet's representation. In addition, it incorporates a separate counting module. Combining these contributions results in a fully automatic method that *outperforms* the state of the art across a wide range of noise levels. We address the reviewers' comments in detail below.

**Reviewer 1**: *Yet, when using neural network one important aspect ... authors should mention training time and energy consumption (or CPU number/GPU type).* This is a good point. We will report training time (11 hours on a NVIDIA 1080Ti), and the corresponding energy consumption. We will highlight that learning-based methodology introduces a new trade-off in signal-processing applications: it requires prior training, but provides fast test-time performance.

**Reviewer 2**: *One of the novelty of the approach is the component counting module : I would have liked to see the influence of this module in terms of performances : what if this module is removed ? Are the improvement due to this module or to the change in the encoder architecture ?* We agree with the reviewer that this is an important point that should be clarified. The proposed frequency-representation and frequency-counting modules are trained separately. Section 3.2 provides an evaluation of the frequency-representation module (and a comparison to other frequency representations). To decouple the evaluation from the performance of the frequency-counting module we assume knowledge of the true number of frequencies. The results show that the novel elements in the architecture of the frequency-representation module produce a significant boost in performance. We will edit the paper to make this clearer.

*What if the number of components is misevaluated ? Does it degrade the results ?* This is a very interesting point. Our framework is robust to mistakes in frequency counting because the frequency-representation module is trained separately from the counting module. This follows from the results in Section 3.2 which show that the $m$ highest local maxima of the learned representation are accurate estimates of the true frequencies ($m$ is the true number of frequencies). As a result, if the counting module wrongly estimates that there are $m-1$ frequencies (for example), those $m-1$ frequencies will be estimated accurately. We will edit the paper to highlight this point.

*Same questions about the noise level : what if this parameter is misevaluated ? Section 3.4 gives some insights about this, but I think that the fact that the method is fully automatic is an important point that would have needed a more detailed discussion.* We will add a detailed discussion about this. In our framework the deep neural net is trained simultaneously over a wide range of noise levels (up to the point where the signal and the noise have the same energy), so there is no need for an explicit estimate of the noise level. This is the case for the results reported in Sections 3.3 and 3.4, which therefore showcase the robustness of the method to variations in the noise.

*How would the standard PSnet behave with an additional component counting module? Would it achieve the same performances?* This is a very good point. The performance of the counting module is highly dependent on the quality of the frequency representation provided as input. The figures below– which are modified versions of Figures 6 and 7 that will be added to the paper– illustrate the performance of PSnet combined with a counting module. The counting module has the same architecture as in DeepFreq, but is trained from scratch using the representation produced by PSnet. Using the PSnet representation results in a significant degradation of performance in both estimation error and Chamfer error.

(a) Figure 6 adding PSnet and a counter module.     (b) Figure 7 adding PSnet and a counter module.

**Reviewer 3**: *For the sinusoid counting task, the error metric is not described ...* We completely agree that this was not clear. The metrics used to train and evaluate the counting module are different. In Section 3.3 the error is computed by counting the fraction of signals in the test set for which the number of components is not estimated correctly. Training is indeed performed by regression, using the squared difference between the true number of frequencies and the non-rounded output of the counting module (as stated briefly at the end of Section 2.3[1]).

## Footnotes

[1]There is a small typo, which we will fix: the cost function is the squared $\ell_2$ norm of the difference not the $\ell_2$ norm.


[Meta-Review · NeurIPS 2019]

This paper describes estimating signals that are sparse superpositions of sinusoids. Overall it is clear and with strong empirical results. However, given how much it builds upon the past work in [23], I don't think it meets the standard for an oral presentation.